# Convergence Analysis of a Momentum Algorithm with Adaptive Step Size for Nonconvex Optimization

## Abstract

Although ADAM is a very popular algorithm for optimizing the weights of neural networks, it has been recently shown that it can diverge even in simple convex optimization examples. Several variants of ADAM have been proposed to circumvent this convergence issue. In this work, we study the ADAM algorithm for smooth nonconvex optimization under a boundedness assumption on the adaptive learning rate. The bound on the adaptive step size depends on the Lipschitz constant of the gradient of the objective function and provides safe theoretical adaptive step sizes. Under this boundedness assumption, we show a novel first order convergence rate result in both deterministic and stochastic contexts. Furthermore, we establish convergence rates of the function value sequence using the Kurdyka-Łojasiewicz property.

## 1 Introduction

Consider the unconstrained optimization problem

$$\min_{x \in \mathbb{R}^d} f(x), \tag{1}$$

where $f : \mathbb{R}^d \to \mathbb{R}$ is a differentiable map and $d$ is an integer. Gradient descent is one of the most classical algorithms to solve this problem. Since the seminal work Robbins and Monro (1951), its stochastic counterpart became one of the most popular algorithms to solve machine learning problems (see Bottou et al. (2018) for a recent survey). Recently, a class of algorithms called adaptive algorithms which are variants of stochastic gradient descent became very popular in machine learning applications. Using a coordinate-wise step size computed using past gradient information, the step size is adapted to the function to optimize and does not follow a predetermined step size schedule. Among these adaptive algorithms, ADAM (Kingma and Ba, 2015) is very popular for optimizing the weights of neural networks. However, recently, Reddi et al. (2018) exhibited a simple convex stochastic optimization problem over a compact set where ADAM fails to converge because of its short-term gradient memory. Moreover, they proposed an algorithm called AMSGRAD to fix the convergence issue of ADAM. This work opened the way to the emergence of other variants of ADAM to overcome its convergence issues (see Section 3 for a detailed review). In this work, under a bounded step size assumption, we propose a theoretical analysis of ADAM for nonconvex optimization.

**Contributions.**

- We establish a convergence rate for ADAM in the deterministic case for nonconvex optimization under a bounded step size. This algorithm can be seen as a deterministic clipped version of ADAM which guarantees safe theoretical step sizes. More precisely, we show a $O(1/n)$ convergence rate by introducing a suitable Lyapunov function.

- We show a similar convergence result to Zaheer et al. (2018, Thm. 1) for nonconvex stochastic optimization up to the limit of the variance of stochastic gradients under an almost surely bounded step size. In comparison to the literature, we relax the hypothesis of the boundedness of the gradients. We also improve the dependency of the convergence result on the dimension $d$ of the parameters.

- We propose a convergence rate analysis of the objective function of the algorithm using the Kurdyka-Łojasiewicz (KŁ) property. To the best of our knowledge, this is the first time such a result is established for an adaptive optimization algorithm.

## 2 A MOMENTUM ALGORITHM WITH ADAPTIVE STEP SIZE

**Notations.** All operations between vectors of $\mathbb{R}^d$ are to read coordinatewise. The vector of ones of $\mathbb{R}^d$ is denoted by $\mathbf{1}$. When a scalar is added to a vector, it is added to each one of its coordinates. Inequalities are also to be read coordinatewise. If $x \in \mathbb{R}^d, x \leq \lambda \in \mathbb{R}$ means that each coordinate of $x$ is smaller than $\lambda$.

We investigate the following algorithm defined by two sequences $(x_n)$ and $(p_n)$ in $\mathbb{R}^d$:

$$\begin{cases} x_{n+1} = x_n - a_{n+1}p_{n+1} \\ p_{n+1} = p_n + b\left(\nabla f(x_n) - p_n\right) \end{cases} \tag{2}$$

where $\nabla f(x)$ is the gradient of $f$ at point $x$, $(a_n)$ is a sequence of vectors in $\mathbb{R}^d$ with positive coordinates, $b$ is a positive real constant and $x_0, p_0 \in \mathbb{R}^d$.

Algorithm (2) includes the classical Heavy-ball method as a special case, but is much more general. Indeed, we allow the sequence of step sizes $(a_n)$ to be adaptive : $a_n \in \mathbb{R}^d$ may depend on the past gradients $g_k := \nabla f(x_k)$ and the iterates $x_k$ for $k \leq n$. We stress that the step size $a_n$ is a vector of $\mathbb{R}^d$ and that the product $a_{n+1}p_{n+1}$ in (2) is read componentwise (this is equivalent to the formulation with a diagonal matrix preconditioner applied to the gradient (McMahan and Streeter, 2010; Gupta et al., 2017; Agarwal et al., 2019)). We present in the following table how to recover some of the famous algorithms with a vector step size formulation:

| Algorithm | Effective step size $a_{n+1}$ | Momentum |
|---|---|---|
| SGD (Robbins and Monro, 1985) | $a_{n+1} \equiv a$ | $b = 1$ (no momentum) |
| ADAGRAD (Duchi et al., 2011) | $a_{n+1} = a\left(\sum_{i=0}^n g_i^2\right)^{-1/2}$ | $b = 1$ |
| RMSPROP (Tieleman and Hinton, 2012) | $a_{n+1} = a\left[\epsilon + \left(c\sum_{i=0}^n(1-c)^{n-i}g_i^2\right)^{1/2}\right]^{-1}$ | $b = 1$ |
| ADAM (Kingma and Ba, 2015) | $a_{n+1} = a\left[\epsilon + \left(c\sum_{i=0}^n(1-c)^{n-i}g_i^2\right)^{1/2}\right]^{-1}$ | $0 \leq b \leq 1$ (close to 0) |

In particular, ADAM (Kingma and Ba, 2015) defined by the iterates :

$$\begin{cases} x_{n+1} = x_n - \frac{a}{\epsilon + \sqrt{v_{n+1}}}p_{n+1} \\ p_{n+1} = p_n + b\left(\nabla f(x_n) - p_n\right) \\ v_{n+1} = v_n + c\left(\nabla f(x_n)^2 - v_n\right) \end{cases} \tag{3}$$

for constants $a \in \mathbb{R}_+$, $b, c \in [0, 1]$, can be seen as an instance of this algorithm by setting $a_n = \frac{a}{\epsilon + \sqrt{v_n}}$ where the vector $v_n$, as defined above, is an exponential moving average of the gradient squared. For simplification, we omit bias correction steps for $p_{n+1}$ and $v_{n+1}$.

We introduce the main assumption on the objective function which is standard in gradient-based algorithms analysis.

**Assumption 2.1.** The mapping $f : \mathbb{R}^d \to \mathbb{R}$ satisfies the following

*(i)* $f$ is continuously differentiable and its gradient $\nabla f$ is $L-$Lipschitz continuous,
*(ii)* $f$ is bounded from below, i.e., $\inf_{x \in \mathbb{R}^d} f(x) > -\infty$.

## 3 RELATED WORKS

### 3.1 THE HEAVY-BALL ALGORITHM.

**Adaptive algorithms as Heavy Ball.** Thanks to its small per-iteration cost and its acceleration properties (at least in the strongly convex case), the Heavy-ball method, also called gradient descent

with momentum, recently regained popularity in large-scale optimization (Sutskever et al., 2013). This speeding up idea dates back to the sixties with the seminal work of Polyak (1964). In order to tackle nonconvex optimization problems, Ochs et al. (2014) proposed iPiano, a generalization of the well known heavy-ball in the form of a forward-backward splitting algorithm with an inertial force for the sum of a smooth possibly nonconvex and a convex function. In the particular case of the Heavy-ball method, this algorithm writes for two sequences of reals $(\alpha_n)$ and $(\beta_n)$:

$$x_{n+1} = x_n - \alpha_n \nabla f(x_n) + \beta_n(x_n - x_{n-1}). \tag{4}$$

We remark that Algorithm (2) can be written in a similar fashion by choosing step sizes $\alpha_n = ba_{n+1}$ and inertial parameters $\beta_n = (1 - b)a_{n+1}/a_n$. Ochs et al. (2014) only consider the case where $\alpha_n$ and $\beta_n$ are real-valued. Moreover, the latter does not consider adaptive step sizes, i.e step sizes depending on past gradient information. We can show some improvement with respect to Ochs et al. (2014) with weaker convergence conditions in terms of the step size of the algorithm (see Appendix B.5) while allowing adaptive vector-valued step sizes $a_n$ (see Proposition A.1).

It is shown in Ochs et al. (2014) that the sequence of function values converges and that every limit point is a critical point of the objective function. Moreover, supposing that the Lyapunov function has the KŁ property at a cluster point, they show the finite length of the sequence of iterates and its global convergence to a critical point of the objective function. Similar results are shown in Wu and Li (2019) for a more general version than iPiano (Ochs et al., 2014) computing gradients at an extrapolated iterate like in Nesterov's acceleration.

**Convergence rate.** Ochs et al. (2014) determines a $O(1/n)$ convergence rate (where $n$ is the number of iterations of the algorithm) with respect to the proximal residual which boils down to the gradient for noncomposite optimization. Furthermore, a recent work introduces a generalization of the Heavy-ball method (and Nesterov's acceleration) to constrained convex optimization in Banach spaces and provides a non-asymptotic hamiltonian based analysis with $O(1/n)$ convergence rate (Diakonikolas and Jordan, 2019). In the same vein, in Section 4, we establish a similar convergence result for an adaptive step size instead of a fixed predetermined step size schedule like in the Heavy-ball algorithm (see Theorem 4.2).

**Convergence rates under the Kurdyka-Łojasiewicz property.** The KŁ property is a powerful tool to analyze gradient-like methods. We elaborate on this property in Section 5. It is for example possible to derive convergence rates assuming that the objective function satisfies this geometric property. Indeed, some recent progress has been made to study convergence rates of the Heavy-ball algorithm in the nonconvex setting. Ochs (2018) establishes local convergence rates for the iterates and the function values sequences under the KŁ property. The convergence proof follows a general method that is often used in non-convex optimization convergence theory. This framework was used for gradient descent (Absil et al., 2005), for proximal gradient descent (see Attouch and Bolte (2009) for an analysis with the Łojasiewicz inequality) and further generalized to a class of descent methods called *gradient-like descent* algorithms (Attouch et al., 2013)(see also for ex. Bolte et al. (2018, Appendix)). KŁ-based asymptotic convergence rates were established for constant Heavy-ball parameters (Ochs, 2018). Asymptotic convergence rates based on the KŁ property were also shown (Johnstone and Moulin, 2017) for a general algorithm solving nonconvex nonsmooth optimization problems called Multi-step Inertial Forward-Backward splitting (Liang et al., 2016) which has iPiano and Heavy-ball methods as special cases. In this work, step sizes and momentum parameter vary along the algorithm run and are not supposed constant. However, specific values are chosen and consequently, their analysis does not encompass adaptive step sizes i.e. stepsizes that can possibly depend on past gradient information. In the present work, we establish similar convergence rates for methods such as ADAM under a bounded step size assumption (see Theorem 5.3). We also mention Li et al. (2017) which analyzes the accelerated proximal gradient method for nonconvex programming (APGnc) and establishes convergence rates of the function value sequence by exploiting the KŁ property. This algorithm is a descent method i.e. the function value sequence is shown to decrease over time. In the present work, we analyze adaptive algorithms which are not descent methods. Note that even Heavy-ball is not a descent method. Hence, our analysis requires additional treatments to exploit the KŁ property : we introduce a suitable Lyapunov function which is not the objective function.

## 3.2 VARIANTS OF ADAM

**Theoretical guarantees of variants of ADAM.** We list most of the existing variants of the ADAM algorithms together with their theoretical convergence guarantees in the following table. The gradient is supposed $L$-lipschitz continuous in all the convergence results. $g_{1:T,i} = [g_{1,i}, g_{2,i}, \cdots, g_{T,i}]^T$.

| Algorithm | Effective step size $a_{n+1}$ | $b_n$ | $c_n$ | Assumptions | Convergence Result |
|---|---|---|---|---|---|
| AMSGRAD[1], ADAMNC[2] (Reddi et al., 2018) | $\frac{a_0}{\sqrt{n}}\frac{1}{\sqrt{\hat{v}_n}}$ 
 $(1)\,\hat{v}_{n+1} = \max(\hat{v}_n, (1-c_n)v_n + c_n g_n^2)$ 
 $(2)\,\hat{v}_{n+1} = (1-c_n)\hat{v}_n + c_n g_n^2$ | $1 - b_1\lambda^{n-1}$ 
 or $1 - \frac{b_1}{n}$ | $\frac{c_1}{n}$ (for ADAMNC) 
 $c_n \equiv c_1$ | • convex functions 
 • bounded gradients 
 • bounded feasible set 
 • $\sum_{i=1}^d \hat{v}_{T,i}^{1/2} \leq d$ (AMSGRAD) 
 • $\sum_{i=1}^d \|g_{1:T,i}\|_2 \leq \sqrt{dT}$ 
 • $b_1 < \sqrt{c_1}$ (ADAMNC) | $R_T/T = O(\sqrt{\log T/T})$ 
 $\frac{R_T}{T} = O(1/\sqrt{T})$ (ADAMNC) |
| ADAM (De et al., 2018) | $\frac{4\|g_n\|^2\eta}{3L(1-(1-b)^n)^2(\eta+2\sigma)^2}\frac{1}{\epsilon+\sqrt{v_n}}$ 
 $v_{n+1} = (1-c_1)v_n + c_1 g_n^2$ | $b_n \equiv b_1$ 
 $= 1 - \frac{\eta}{n}$ | $c_n \equiv c_1$ | • $\sigma$-bounded gradients 
 • $\epsilon = 2\sigma$ | $\forall \eta > 0 \; \exists n \leq \frac{9L\sigma^2(f(x_2)-f(x_*))}{\eta^6}$ 
 s.t. $\|g_n\| \leq \eta$ |
| PADAM, AMSGRAD (Zhou et al., 2018) | $\frac{1}{\sqrt{N}}\frac{1}{\hat{v}_n^p}$ (AMSGRAD) 
 $\hat{v}_n = \max(\hat{v}_{n-1}, (1-c)v_{n-1} + cg_n^2)$ | $b_n \equiv b$ | $c_n \equiv c$ | • bounded gradients 
 For PADAM: • $p \in [0, \frac{1}{4}]$ 
 • $1 - b < (1-c)^{2p}$ 
 • $\sum_{i=1}^d \|g_{1:N,i}\|_2 \leq \sqrt{dN}$ 
 AMSGRAD: $p = \frac{1}{2}$ and $1 - b < 1 - c$ | $\mathbb{E}[\|g_\tau\|^2] = O(\frac{1+\sqrt{d}}{\sqrt{N}} + \frac{d}{N})$ 
 $= O(\sqrt{\frac{d}{N}} + \frac{d}{N})$ (AMSGRAD) 
 $\tau$ uniform r.v in $\{1, \cdots, N\}$ |
| RMSPROP[1], YOGI[2] (Zaheer et al., 2018) | $\frac{a_1}{\epsilon+\sqrt{v_n}}$ 
 $(1)\,v_{n+1} = (1-c_n)v_n + cg_n^2$ 
 $(2)\,v_n = v_{n-1} - (1-c)\text{sign}(v_{n-1}-g_n^2)$ | $b_n \equiv b$ | $c_n \equiv c$ | • $G$-bounded gradients 
 • $a_1 \leq \frac{\epsilon\sqrt{1-c}}{2L}$ (YOGI) 
 • $a_1 \leq \frac{\epsilon}{2L}$ • $c \leq \frac{\epsilon^2}{16G^2}$ 
 • $\sigma^2$-bounded variance | $\mathbb{E}[\|g_\tau\|^2] = O(\frac{1}{N} + \sigma^2)$ 
 $\tau$ uniform r.v in $\{1, \cdots, N\}$ 
 $O(\frac{1}{N})$ if minibatch $\Theta(N)$ |
| AMSGRAD[1], ADAFOM[2] (Chen et al., 2019) | $\frac{1}{\sqrt{n}}\frac{1}{\sqrt{\hat{v}_n}}$ 
 $(1)\,\hat{v}_{n+1} = \max(\hat{v}_n, (1-c_n)v_n + c_n g_n^2)$ 
 $(2)\,\hat{v}_{n+1} = (1-\frac{1}{n})\hat{v}_n + \frac{1}{n}g_n^2$ | non-increasing | $c_n \equiv c_1$ | • bounded gradients 
 • $\exists c > 0$ s.t. $|g_{1,i}| \geq c$ | $\min_{n\in[0,N]} \mathbb{E}[\|g_n\|^2] = O(\frac{\log N + d^2}{\sqrt{N}})$ |
| GENERIC ADAM (Zou et al., 2019) | $\frac{\alpha_n}{\sqrt{v_n}}$ 
 $v_{n+1} = (1-c_n)v_n + c_n g_n^2$ 
 $\alpha_n = \hat{\alpha}\frac{\sqrt{1-(1-c)^n}}{1-(1-b)^n}$ | $b_n \geq b > 0$ | $0 < c_n < 1$ 
 non-increasing 
 $\lim c_n = c > b^2$ | • bounded gradients in expectation 
 • $d_n \leq \frac{\alpha_n}{\sqrt{c_n}} \leq c_0 d_n$ 
 $d_n$ non-increasing | $\mathbb{E}[\|g_\tau\|^{\frac{4}{3}}]^{\frac{3}{2}} \leq \frac{C+C'\sum_{n=1}^N \alpha_n\sqrt{c_n}}{N\alpha_N}$ 
 $\tau$ uniform r.v in $\{1, \cdots, N\}$ |
| ADABOUND[1], AMSBOUND[2] (Luo et al., 2019) | $\frac{1}{\sqrt{n}}\text{clip}(\frac{\alpha}{\sqrt{v_n}}, \eta_l(n), \eta_u(n))$ 
 $\eta_l(n)$ non-decreasing to $\alpha_*$ 
 $\eta_u(n)$ non-increasing to $\alpha_*$ 
 $(1)\,v_{n+1} = (1-c)v_n + cg_n^2$ 
 $(2)\,v_{n+1} = \max(v_n, (1-c)v_n + cg_n^2)$ | $1 - (1-b)\lambda^{n-1}$ 
 or $1 - \frac{b}{1-b}\frac{n}{}$ 
 $b_n \geq b$ | $c_n \equiv c$ | • bounded gradients 
 • closed convex 
 bounded feasible set 
 • $1 - b < \sqrt{1-c}$ | $R_T/T = O(1/\sqrt{T})$ |

**Discussion of theoretical results.** The first type of convergence results uses the online optimization framework which controls the convergence rate of the average regret. This framework was adopted for AMSGRAD, ADAMNC (Reddi et al., 2018), ADABOUND and AMSBOUND (Luo et al., 2019). In this setting, it is assumed that the feasible set containing the iterates is bounded by adding a projection step to the algorithm if needed. We do not make such an assumption in our analysis. (Reddi et al., 2018) establishes a regret bound in the convex setting. The second type of theoretical results is based on the control of the norm of the (stochastic) gradients. We remark that some of these results depend on the dimension of the parameters. Zhou et al. (2018) improves this dependency in comparison to Chen et al. (2019). The convergence result in De et al. (2018) is established under quite specific values of $a_{n+1}, b_n$ and $\epsilon$. Zaheer et al. (2018) show a $O(1/n)$ convergence rate for an increasing mini-batch size. However, the proof is provided for RMSPROP and seems difficult to adapt to ADAM which involves a momentum term. Indeed, unlike RMSPROP, ADAM does not admit the objective function as a Lyapunov function. Although we assume boundedness of the step size by Condition (7), we do not suppose that $a_1 \leq \frac{\varepsilon}{2L}$ (see table in Section 3.2) which can impose a very small step size and result in a slow convergence. The step size assumption $a_1 \leq \frac{\epsilon}{2L}$ imposes a very small step size which may result in a slow convergence. We also remark that all the available theoretical results assume boundedness of the (stochastic) gradients. We do not make such an assumption. Furthermore, we do not add any decreasing $1/\sqrt{n}$ factor in front of the adaptive step size as it is considered in Reddi et al. (2018); Luo et al. (2019) and Chen et al. (2019). Although constant hyperparameters $b$ and $c$ are used in practice, theoretical results are often established for non constant $b_n$ and $c_n$ (Reddi et al., 2018; Luo et al., 2019). We also mention that most of the theoretical bounds depend on the dimension of the parameter (Reddi et al., 2018; Zhou et al., 2018; Chen et al., 2019; Zou et al., 2019; Luo et al., 2019).

**Other variants of ADAM.** Recently, several other algorithms were proposed in the literature to enhance ADAM. Although these algorithms lack theoretical guarantees, they present interesting ideas and show good practical performance. For instance, ADASHIFT (Zhou et al., 2019) argues that the convergence issue of ADAM is due to its unbalanced step sizes. To solve this issue, they propose to use temporally shifted gradients to compute the second moment estimate in order to decorrelate it from the first moment estimate. NADAM (Dozat, 2016) incorporates Nesterov's acceleration into ADAM in order to improve its speed of convergence. Moreover, originally motivated by variance reduction, QHADAM (Ma and Yarats, 2019) replaces both ADAM's moment estimates by quasi-hyperbolic terms and recovers ADAM, RMSPROP and NADAM as particular cases (modulo the bias correction). Guided by the same variance reduction principle, RADAM (Liu et al., 2019) estimates the variance of the effective step size of the algorithm and proposes a multiplicative variance correction to the update rule.

**Step size bound.** Perhaps, the closest idea to our algorithm is the recent ADABOUND (Luo et al., 2019) which considers a dynamic learning rate bound. Luo et al. (2019) show that extremely small and large learning rates can cause convergence issues to ADAM and exhibit empirical situations where such an issue shows up. Inspired by the gradient clipping strategy proposed in Pascanu et al. (2013) to tackle the problem of vanishing and exploding gradients in training recurrent neural networks, Luo et al. (2019) apply clipping to the effective step size of the algorithm in order to circumvent step size instability. More precisely, authors propose dynamic bounds on the learning rate of adaptive methods such as ADAM or AMSGRAD to solve the problem of extreme learning rates which can lead to poor performance. Initialized respectively at 0 and ∞, lower and upper bounds both converge smoothly to a constant final step size following a predetermined formula defined by the user. Consequently, the algorithm resembles an adaptive algorithm in the first iterations and becomes progressively similar to a standard SGD algorithm. Our approach is different : we propose a static bound on the adaptive learning rate which depends on the Lipschitz constant of the objective function. This bound stems naturally from our theoretical derivations.

# 4 FIRST ORDER CONVERGENCE RATE

## 4.1 DETERMINISTIC SETTING

Let $(H_n)_{n\geq 0}$ be a sequence defined for all $n \in \mathbb{N}$ by $H_n := f(x_n) + \frac{1}{2b}\langle a_n, p_n^2 \rangle$.

We further assume the following step size growth condition.

**Assumption 4.1.** There exists $\alpha > 0$ s.t. $a_{n+1} \leq \frac{a_n}{\alpha}$.

Note that this assumption is satisfied for ADAM with $\alpha = \sqrt{1-c}$ where $c$ is the parameter in (3). Unlike in AMSGRAD (Reddi et al., 2018), the step size is not necessarily nonincreasing.

We provide a proof of the following key lemma in Appendix A.1.

**Lemma 4.1.** Let Assumptions 2.1 and 4.1 hold true. Then, for all $n \in \mathbb{N}$, for all $u \in \mathbb{R}_+$,

$$H_{n+1} \leq H_n - \langle a_{n+1} p_{n+1}^2, A_{n+1} \rangle - \frac{b}{2} \langle a_{n+1}(\nabla f(x_n) - p_n)^2, B\mathbf{1} \rangle, \qquad (5)$$

where $A_{n+1} := 1 - \frac{a_{n+1}L}{2} - \frac{|b - (1-\alpha)|}{2u} - \frac{1-\alpha}{2b}$ and $B := 1 - \frac{|b - (1-\alpha)|u}{b} - (1-\alpha)$

We state now one of the principal convergence results about Algorithm 2. In particular, we establish a sublinear convergence rate for the minimum of the gradients norms until time $n$.

**Theorem 4.2.** Let Assumptions 2.1 and 4.1 hold true. Suppose that $1 - \alpha < b \leq 1$. Let $\varepsilon > 0$ s.t. $a_{\sup} := \frac{2}{L} \left( 1 - \frac{(b-(1-\alpha))^2}{2b\alpha} - \frac{1-\alpha}{2b} - \varepsilon \right)$ is nonnegative. Let $\delta > 0$ s.t. for all $n \in \mathbb{N}$,

$$\delta \leq a_{n+1} \leq \min \left( a_{\sup}, \frac{a_n}{\alpha} \right). \qquad (6)$$

Then,

(i) the sequence $(H_n)$ is nonincreasing and $\sum_n \|p_n\|^2 < \infty$.
In particular, $\lim x_{n+1} - x_n \to 0$ and $\lim \nabla f(x_n) \to 0$ as $n \to +\infty$.

(ii) For all $n \geq 1$, $\quad \min_{0 \leq k \leq n-1} \|\nabla f(x_k)\|^2 \leq \frac{4}{nb^2} \left( \frac{H_0 - \inf f}{\delta \varepsilon} + \|p_0\|^2 \right)$.

We provide some comments on this result.

**Dimension dependence.** Unlike most of the theoretical results for variants of ADAM as gathered in Section 3.2, we remark that the bound (ii) does not depend on the dimension $d$ of the parameter $x_k$.

**Comparison to gradient descent.** A similar result holds for deterministic gradient descent. If $\gamma$ is a fix step size for gradient descent and there exist $\delta > 0, \varepsilon > 0$ s.t. $\gamma > \delta$ and $1 - \frac{\gamma L}{2} > \varepsilon$, then ( see Appendix B.6) for all $n \geq 1$:

$$\min_{0 \leq k \leq n-1} \|\nabla f(x_k)\|^2 \leq \frac{f(x_0) - \inf f}{n\gamma(1 - \frac{\gamma L}{2})} \leq \frac{f(x_0) - \inf f}{n\delta \varepsilon}.$$

When $p_0 = 0$ (this is the case for ADAM ), the bound in Theorem 4.2 coincides with the gradient descent bound, up to the constant $4/b^2$. We mention however that $\varepsilon$ for Algorithm (2) is defined by a slightly more restrictive condition than for gradient descent : when $b = 1$, there is no momentum and $a_{\sup} = \frac{1}{L}(1 - 2\varepsilon) < 2/L$. Hence, under the boundedness of the effective step size, the algorithm have a similar convergence guarantee to gradient descent. Remark that the step size bound almost matches the classical $2/L$ upperbound on the step size of gradient descent. As it is already known for gradient descent, a large step size, even if it is adaptive, can harm the convergence of the algorithm.

## 4.2 STOCHASTIC SETTING

We establish a similar bound in the stochastic setting. Let $(\Xi, \mathfrak{S})$ denote a measurable space and $d \in \mathbb{N}$. Consider the problem of finding a local minimizer of the expectation $F(x) := \mathbb{E}(f(x, \xi))$ w.r.t. $x \in \mathbb{R}^d$, where $f : \mathbb{R}^d \times \Xi \to \mathbb{R}$ is a measurable map and $f(\,.\,, \xi)$ is a possibly nonconvex function depending on some random variable $\xi$. The distribution of $\xi$ is assumed unknown, but revealed online by the observation of iid copies $(\xi_n : n \geq 1)$ of the r.v. $\xi$. For a fixed value of $\xi$, the mapping $x \mapsto f(x, \xi)$ is supposed to be differentiable, and its gradient w.r.t. $x$ is denoted by $\nabla f(x, \xi)$. We study a stochastic version of Algorithm (2) by replacing the deterministic gradient $\nabla f(x_n)$ by $\nabla f(x_n, \xi_{n+1})$.

**Theorem 4.3.** Let Assumption 2.1 (for $F$) and Assumption 4.1 hold true. Assume the following bound on the variance in stochastic gradients: $\mathbb{E}\|\nabla f(x, \xi) - \nabla F(x)\|^2 \leq \sigma^2$ for all $x \in \mathbb{R}^d$.

Suppose moreover that $1 - \alpha < b \leq 1$. Let $\varepsilon > 0$ s.t. $\bar{a}_{\sup} := \frac{1}{L} \left( 1 - \frac{(b-(1-\alpha))^2}{b\alpha} - \frac{1-\alpha}{b} - \varepsilon \right)$ is nonnegative. Let $\delta > 0$ s.t. for all $n \geq 1$, almost surely,

$$\delta \leq a_{n+1} \leq \min \left( \bar{a}_{\sup}, \frac{a_n}{\alpha} \right) . \tag{7}$$

Then,

$$\mathbb{E}[\|\nabla F(x_\tau)\|^2] \leq \frac{4}{n\delta b^2 \alpha} \left( \frac{H_0 - \inf f}{\varepsilon} + \|\sqrt{a_0} p_0\|^2 + \frac{n\bar{a}_{\sup}\sigma^2}{2\varepsilon} \right) ,$$

where $x_\tau$ is an iterate uniformly randomly chosen from $\{x_0, \cdots, x_{n-1}\}$.

The proof is defered to the appendix. In the special case where there is no momentum in the algorithm (i.e. RMSPROP) and assuming that the gradients are bounded, a similar convergence rate is obtained in Zaheer et al. (2018, Thm. 1) (see Section 3.2).

## 5 CONVERGENCE RATE ANALYSIS UNDER THE KŁ PROPERTY

The KŁ inequality has been used to show the convergence of several first-order optimization methods towards critical points (Attouch and Bolte, 2009; Attouch et al., 2010; 2013; Bolte et al., 2014; Frankel et al., 2015; Li et al., 2017). In this section, we use a methodology exposed in Bolte et al. (2018, Appendix) to show convergence rates based on the KŁ property. We modify it to encompass momentum methods. Note that although this modification was initiated in Ochs et al. (2014); Ochs (2018), we use a different separable Lyapunov function. The first part of the proof follows these approaches and the second part follows the proof of Johnstone and Moulin (2017, Theorem 2).

Consider the function $H : \mathbb{R}^d \times \mathbb{R}^d \to \mathbb{R}$ defined for all $z = (x, y) \in \mathbb{R}^d \times \mathbb{R}^d$ by

$$H(z) = H(x, y) = f(x) + \frac{1}{2b}\|y\|^2 . \tag{8}$$

Notice that $H_n = f(x_n) + \frac{1}{2b}\langle a_n, p_n^2 \rangle = H(x_n, y_n)$ where $(y_n)_{n \in \mathbb{N}}$ is a sequence defined for all $n \in \mathbb{N}$ by $y_n = \sqrt{a_n} p_n$.

**Notations and definitions.** If $(E, \mathsf{d})$ is a metric space, $z \in E$ and $A$ is a non-empty subset of $E$, we use the notation $\mathsf{d}(z, A) := \inf\{\mathsf{d}(z, z') : z' \in A\}$. The set of critical points of the function $H$ is defined by $\operatorname{crit} H := \{z \in \mathbb{R}^{2d} \text{ s.t. } \nabla H(z) = 0\}$.

**Definition 5.1.** (Set of limit points) The set of all limit points of $(z_k)_{k \in \mathbb{N}}$ initialized at $z_0$ is defined by $\omega(z_0) := \{\bar{z} \in \mathbb{R}^{2d} : \exists \text{ an increasing sequence of integers } (k_j)_{j \in \mathbb{N}} \text{ s.t } z_{k_j} \to \bar{z} \text{ as } j \to \infty\}$.

**Assumption 5.1.** $f$ is coercive.

**Lemma 5.1.** (Properties of the limit point set) Let $(z_k)_{k \in \mathbb{N}}$ be the sequence defined for all $k \in \mathbb{N}$ by $z_k = (x_k, y_k)$ where $y_k = \sqrt{a_k} p_k$ and $(x_k, p_k)$ is generated by Algorithm (2) from a starting point $z_0$. Let Assumptions 2.1, 4.1 and 5.1 hold true. Assume that Condition (6) holds. Then,

    (i) $\omega(z_0)$ is a nonempty compact set.
    (ii) $\omega(z_0) \subset \operatorname{crit} H = \operatorname{crit} f \times \{0\}$.
    (iii) $\lim\limits_{k \to +\infty} \mathsf{d}(z_k, \omega(z_0)) = 0$.
    (iv) $H$ is finite and constant on $\omega(z_0)$.

We introduce the KŁ inequality which is the key tool of our analysis. We refer to Bolte et al. (2010) for an in-depth presentation of the KŁ property historically introduced by the fundamental works of Łojasiewicz (1963) and Kurdyka (1998).

Define $[\alpha < H < \beta] := \{z \in \mathbb{R}^{2d} : \alpha < H(z) < \beta\}$. Let $\eta > 0$ and define

$$\Phi_\eta := \{\varphi \in C^0[0, \eta) \cap C^1(0, \eta) : \varphi(0) = 0, \varphi \text{ concave and } \varphi' > 0\} .$$

where $C^0[0, \eta)$ is the set of continuous functions on $[0, \eta)$ and $C^1(0, \eta)$ is the set of continuously differentiable functions on $(0, \eta)$.

**Definition 5.2.** (KŁ property, Bolte et al. (2018, Appendix)) A proper and lower semicontinuous (l.s.c) function $H : \mathbb{R}^{2d} \to (-\infty, +\infty]$ has the KŁ property locally at $\bar{z} \in \operatorname{dom} H$ if there exist $\eta > 0$, $\varphi \in \Phi_\eta$ and a neighborhood $U(\bar{z})$ s.t. for all $z \in U(\bar{z}) \cap [H(\bar{z}) < H < H(\bar{z}) + \eta]$:

$$\varphi'(H(z) - H(\bar{z})) \|\nabla H(z)\| \geq 1 . \tag{9}$$

When $H(\bar{z}) = 0$, we can rewrite Equation (9) as : $\|\nabla(\varphi \circ H)(z)\| \geq 1$ for suitable $z$ points. This means that $H$ becomes sharp under a reparameterization of its values through the so-called desingularizing function $\varphi$. The function $H$ is said to be a KŁ function if it has the KŁ property at each point of the domain of its gradient. The KŁ property is satisfied by the broad class of semialgebraic functions including most objective functions in real applications ($\| \cdot \|_p$ for $p$ rational, real polynomials, rank, etc.). (see Bolte et al. (2014, Appendix) for more examples). KŁ inequality holds at any non critical point (see Attouch et al. (2010, Remark 3.2 (b))).

We introduce now a uniformized version of the KŁ property which will be useful for our analysis.

**Lemma 5.2.** (Uniformized KŁ property, Bolte et al. (2014, Lemma 6, p 478)) Let $\Omega$ be a compact set and let $H : \mathbb{R}^{2d} \to (-\infty, +\infty]$ be a proper l.s.c function. Assume that $H$ is constant on $\Omega$ and satisfies the KŁ property at each point of $\Omega$. Then, there exist $\varepsilon > 0, \eta > 0$ and $\varphi \in \Phi_\eta$ such that for all $\bar{z} \in \Omega$, for all $z \in \{z \in \mathbb{R}^d : \mathsf{d}(z, \Omega) < \varepsilon\} \cap [H(\bar{z}) < H < H(\bar{z}) + \eta]$, one has

$$\varphi'(H(z) - H(\bar{z}))\|\nabla H(z)\| \geq 1 \tag{10}$$

**Definition 5.3.** (KŁ exponent) If $\varphi$ can be chosen as $\varphi(s) = \frac{\bar{c}}{\theta} s^\theta$ for some $\bar{c} > 0$ and $\theta \in (0, 1]$ in Definition 5.2, then we say that $H$ has the KŁ property at $\bar{z}$ with an exponent of $\theta$ [1]. We say that $H$ is a KŁ function with an exponent $\theta$ if it has the same exponent $\theta$ at any $\bar{z}$.

Furthermore, if $H$ is a proper closed semialgebraic function, then $H$ is a KŁ function with a suitable exponent $\theta \in (0, 1]$. The slope of $\varphi$ around the origin informs about the "flatness" of a function around a point. Hence, the KŁ exponent allows to obtain convergence rates. In the light of this remark, we state one of the main results of this work.

**Theorem 5.3.** (Convergence rates) Let $(z_k)_{k \in \mathbb{N}}$ be the sequence defined for all $k \in \mathbb{N}$ by $z_k = (x_k, y_k)$ where $y_k = \sqrt{a_k} p_k$ and $(x_k, p_k)$ is generated by Algorithm (2) from a starting point $z_0$. Let Assumptions 2.1, 4.1 and 5.1 hold true. Assume that Condition (6) holds. Suppose moreover that $H$ is a KŁ function with KŁ exponent $\theta$. Denote by $f(x_*)$ the limit of the sequence $(H(z_k))_{k \in \mathbb{N}}$ where $x_*$ is a critical point of $f$. Then, the following convergence rates hold:

  (i) If $\theta = 1$, then $f(x_k)$ converges in a finite number of iterations.
 (ii) If $1/2 \leq \theta < 1$, then $f(x_k)$ converges to $f(x_*)$ linearly i.e. there exist $q \in (0, 1), C > 0$
      s.t. $f(x_k) - f(x_*) \leq C q^k$.
(iii) If $0 < \theta < 1/2$, then $f(x_k) - f(x_*) = O(k^{\frac{1}{2\theta - 1}})$.

**Sketch of the proof.** The proof consists of two main steps. The first one is to show that the iterates enter and stay in a region where the KŁ inequality holds. This is achieved using the properties of the limit set (Lemma 5.1) and the uniformized KŁ property (Lemma 5.2). Then, the second step is to exploit this inequality in order to derive the sought convergence results. We defer the complete proof to Appendix B.3.

We introduce a lemma in order to make the KŁ assumption on the objective function $f$ instead of the auxiliary function $H$.

**Lemma 5.4.** Let $f$ be a continuously differentiable function satisfying the KL property at $\bar{x}$ with an exponent of $\theta \in (0, 1/2]$. Then the function $H$ defined in Equation (8) has also the KŁ property at $(\bar{x}, 0)$ with an exponent of $\theta$.

The following result derives a convergence rate on the objective function values under a KŁ assumption on this same function instead of an assumption on the Lyapunov function $H$. The result is an immediate consequence of Lemma 5.4 and Theorem 5.3.

**Corollary 5.5.** Let $(z_k)_{k \in \mathbb{N}}$ be the sequence defined for all $k \in \mathbb{N}$ by $z_k = (x_k, y_k)$ where $y_k = \sqrt{a_k} p_k$ and $(x_k, p_k)$ is generated by Algorithm (2) from a starting point $z_0$. Let Assumptions 2.1, 4.1 and 5.1 hold true. Assume that Condition (6) holds. Suppose moreover that $f$ is a KŁ function with KŁ exponent $\theta \in (0, 1/2)$. Denote by $f(x_*)$ the limit of the sequence $(H(z_k))_{k \in \mathbb{N}}$ where $x_*$ is a critical point of $f$. Then $f(x_k) - f(x_*) = O(k^{\frac{1}{2\theta - 1}})$.

---

[1]$\alpha := 1 - \theta$ is also defined as the KŁ exponent in other papers (Li and Pong, 2018).

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

# A    APPENDIX A

## A.1    PROOF OF LEMMA 4.1

Supposing that $\nabla f$ is $L-$Lipschitz, using Taylor's expansion and the expression of $p_n$ in the algorithm, we obtain the following inequality:

$$f(x_{n+1}) \leq f(x_n) - \langle \nabla f(x_n), a_{n+1} p_{n+1} \rangle + \frac{L}{2} \|a_{n+1} p_{n+1}\|^2 \tag{11}$$

Moreover,

$$\frac{1}{2b} \langle a_{n+1}, p_{n+1}^2 \rangle - \frac{1}{2b} \langle a_n, p_n^2 \rangle = \frac{1}{2b} \langle a_{n+1}, p_{n+1}^2 - p_n^2 \rangle + \frac{1}{2b} \langle a_{n+1} - a_n, p_n^2 \rangle. \tag{12}$$

Observing that $p_{n+1}^2 - p_n^2 = -b^2 (\nabla f(x_n) - p_n)^2 + 2b p_{n+1} (\nabla f(x_n) - p_n)$, we obtain after simplification :

$$H_{n+1} \leq H_n + \frac{L}{2} \|a_{n+1} p_{n+1}\|^2 - \frac{b}{2} \langle a_{n+1}, (\nabla f(x_n) - p_n)^2 \rangle - \langle a_{n+1} p_{n+1}, p_n \rangle + \frac{1}{2b} \langle a_{n+1} - a_n, p_n^2 \rangle. \tag{13}$$

Using again $p_n = p_{n+1} - b(\nabla f(x_n) - p_n)$, we replace $p_n$ :

$$H_{n+1} \leq H_n + \frac{L}{2} \|a_{n+1} p_{n+1}\|^2 - \frac{b}{2} \langle a_{n+1}, (\nabla f(x_n) - p_n)^2 \rangle$$
$$- \langle a_{n+1}, p_{n+1}^2 \rangle + b \langle a_{n+1} p_{n+1}, \nabla f(x_n) - p_n \rangle + \frac{1}{2b} \langle a_{n+1} - a_n, p_n^2 \rangle.$$

Under Assumption 4.1, we write: $\langle a_{n+1} - a_n, p_n^2 \rangle \leq (1 - \alpha) \langle a_{n+1}, p_n^2 \rangle$ and using $p_n^2 = p_{n+1}^2 + b^2 (\nabla f(x_n) - p_n)^2 - 2b p_{n+1} (\nabla f(x_n) - p_n)$, it holds that:

$$H_{n+1} \leq H_n - \langle a_{n+1}, p_{n+1}^2 \rangle - \frac{b}{2} \langle a_{n+1}, (\nabla f(x_n) - p_n)^2 \rangle$$
$$+ \frac{L}{2} \|a_{n+1} p_{n+1}\|^2 + (b - (1 - \alpha)) \langle a_{n+1} p_{n+1}, \nabla f(x_n) - p_n \rangle$$
$$+ \frac{1 - \alpha}{2b} \langle a_{n+1}, p_{n+1}^2 \rangle + \frac{b(1 - \alpha)}{2} \langle a_{n+1}, (\nabla f(x_n) - p_n)^2 \rangle.$$

Using the classical inequality $xy \leq \frac{x^2}{2u} + \frac{uy^2}{2}$, we have :

$$(b - (1 - \alpha)) a_{n+1} p_{n+1} (\nabla f(x_n) - p_n) \leq \frac{|b - (1 - \alpha)|}{2u} \langle a_{n+1}, p_{n+1}^2 \rangle + \frac{|b - (1 - \alpha)| u}{2} \langle a_{n+1}, (\nabla f(x_n) - p_n)^2 \rangle. \tag{14}$$

Hence, after using this inequality and rearranging the terms, we derive the following inequality:

$$
\begin{aligned}
H_{n+1} \leq H_n - \langle a_{n+1} p_{n+1}^2, 1 - \frac{a_{n+1} L}{2} - \frac{|b - (1-\alpha)|}{2u} - \frac{1-\alpha}{2b} \rangle \\
- \frac{b}{2} \langle a_{n+1} (\nabla f(x_n) - p_n)^2, \left( 1 - \frac{|b - (1-\alpha)|u}{b} - (1-\alpha) \right) \mathbf{1} \rangle.
\end{aligned}
$$

This concludes the proof.

## A.2 AN ADDITIONAL PROPOSITION

**Proposition A.1.** Let Assumption 2.1 hold true. Suppose moreover that $1 - \alpha < b \leq 1$. Let $\varepsilon > 0$ s.t. $a_{\sup} := \frac{2}{L} \left( 1 - \frac{(b-(1-\alpha))^2}{2b\alpha} - \frac{1-\alpha}{2b} - \varepsilon \right)$ is nonnegative. Let $\delta > 0$ s.t. for all $n \in \mathbb{N}$,

$$
a_{n+1} \leq \min \left( a_{\sup}, \frac{a_n}{\alpha} \right) .
$$

Then, for all $n \geq 1$,

$$
\sum_{k=0}^{n-1} \langle a_{k+1}, \nabla f(x_k)^2 \rangle \leq \frac{4}{b^2 \alpha} \left( \frac{H_0 - \inf f}{\varepsilon} + \langle a_0, p_0^2 \rangle \right)
$$

*Proof.* This is a consequence of Lemma 4.1. Conditions $A_{n+1} \geq \varepsilon$ and $B \geq 0$ write as follow :

$$
a_{n+1} \leq \frac{2}{L} \left( 1 - \frac{b - (1-\alpha)}{2u} - \frac{1-\alpha}{2b} - \varepsilon \right) \quad \text{and} \quad u \leq \frac{\alpha b}{b - (1-\alpha)} .
$$

We get the assumption made in the proposition by injecting the second condition into the first one and adding the assumption $\frac{a_{n+1}}{a_n} \leq \alpha$ made in the lemma. Under this assumption, we sum over $0 \leq k \leq n - 1$ Equation (5), rearrange it and use $A_{n+1} \geq \varepsilon$, $B \geq 0$ to obtain :

$$
\sum_{k=0}^{n-1} \varepsilon \langle a_{k+1}, p_{k+1}^2 \rangle \leq H_0 - H_n ,
$$

Then, observe that $H_n \geq f(x_n) \geq \inf f$. Therefore, we derive :

$$
\sum_{k=0}^{n-1} \langle a_{k+1}, p_{k+1}^2 \rangle \leq \frac{H_0 - \inf f}{\varepsilon} . \tag{15}
$$

Moreover, from the Algorithm 2 second update rule, we get $\nabla f(x_k) = \frac{1}{b} p_{k+1} - \frac{1-b}{b} p_k$. Hence, we have for all $k \geq 0$ :

$$
\nabla f(x_k)^2 \leq 2 \left( \frac{1}{b^2} p_{k+1}^2 + \frac{(1-b)^2}{b^2} p_k^2 \right) \leq \frac{2}{b^2} (p_{k+1}^2 + p_k^2) .
$$

We deduce that :

$$
\begin{aligned}
\sum_{k=0}^{n-1} \langle a_{k+1}, \nabla f(x_k)^2 \rangle &\leq \frac{2}{b^2} \sum_{k=0}^{n-1} \langle a_{k+1}, p_{k+1}^2 + p_k^2 \rangle \\
&= \frac{2}{b^2} \sum_{k=0}^{n-1} \langle a_{k+1}, p_{k+1}^2 \rangle + \frac{2}{b^2} \sum_{k=0}^{n-1} \langle a_{k+1}, p_k^2 \rangle \\
&\leq \frac{2}{b^2} \sum_{k=0}^{n-1} \langle a_{k+1}, p_{k+1}^2 \rangle + \frac{2}{b^2 \alpha} \sum_{k=0}^{n-1} \langle a_k, p_k^2 \rangle \\
&\leq \frac{2}{b^2 \alpha} \left( 2 \sum_{k=1}^{n-1} \langle a_k, p_k^2 \rangle + \langle a_n, p_n^2 \rangle + \langle a_0, p_0^2 \rangle \right) \\
&\leq \frac{4}{b^2 \alpha} \sum_{k=0}^{n} \langle a_k, p_k^2 \rangle \\
&\leq \frac{4}{b^2 \alpha} \left( \frac{H_0 - \inf f}{\varepsilon} + \langle a_0, p_0^2 \rangle \right) .
\end{aligned}
$$

$\square$

### A.3 Proof of Theorem 4.2

This is a consequence of Lemma 4.1. Conditions $A_{n+1} \geq \varepsilon$ and $B \geq 0$ write as follow :

$$
a_{n+1} \leq \frac{2}{L} \left( 1 - \frac{b - (1-\alpha)}{2u} - \frac{1-\alpha}{2b} - \varepsilon \right) \quad \text{and} \quad u \leq \frac{\alpha b}{b - (1-\alpha)} .
$$

We get the assumption made in the proposition by injecting the second condition into the first one and adding the assumption $\frac{a_{n+1}}{a_n} \leq \alpha$ made in the lemma. Under this assumption, we sum over $0 \leq k \leq n-1$ Equation (5), rearrange it and use $A_{n+1} \geq \varepsilon$, $B \geq 0$ and $a_{k+1} \geq \delta$ to obtain :

$$
\sum_{k=0}^{n-1} \delta \varepsilon \| p_{k+1} \|^2 \leq H_0 - H_n ,
$$

Then, observe that $H_n \geq f(x_n) \geq \inf f$. Therefore, we derive :

$$
\sum_{k=0}^{n-1} \| p_{k+1} \|^2 \leq \frac{H_0 - \inf f}{\delta \varepsilon} . \tag{16}
$$

Moreover, from the algorithm 2 second update rule, we get $\nabla f(x_k) = \frac{1}{b} p_{k+1} - \frac{1-b}{b} p_k$. Hence, we have for all $k \geq 0$ :

$$
\| \nabla f(x_k) \|^2 \leq 2 \left( \frac{1}{b^2} \| p_{k+1} \|^2 + \frac{(1-b)^2}{b^2} \| p_k \|^2 \right) \leq \frac{2}{b^2} ( \| p_{k+1} \|^2 + \| p_k \|^2 ) .
$$

We deduce that :

$$
\sum_{k=0}^{n-1} \| \nabla f(x_k) \|^2 \leq \frac{2}{b^2} \sum_{k=0}^{n-1} (\| p_{k+1} \|^2 + \| p_k \|^2) = \frac{2}{b^2} \left( 2 \sum_{k=1}^{n-1} \| p_k \|^2 + \| p_n \|^2 + \| p_0 \|^2 \right) \leq \frac{4}{b^2} \sum_{k=0}^{n} \| p_k \|^2 .
$$
$$
\tag{17}
$$

Finally, using Equations (16) and (17), we have :

$$
\min_{0 \leq k \leq n-1} \| \nabla f(x_k) \|^2 \leq \frac{1}{n} \sum_{k=0}^{n-1} \| \nabla f(x_k) \|^2 \leq \frac{4}{nb^2} \left( \frac{H_0 - \inf f}{\delta \varepsilon} + \| p_0 \|^2 \right)
$$

### A.4 PROOF OF THEOREM 4.3

The proof of this proposition mainly follows the same path as its deterministic counterpart. However, due to stochasticity, a residual term (the last term in Equation (18)) quantifying the difference between the stochastic gradient estimate and the true gradient of the objective function (compare Equation (18) to Lemma 4.1). Following the exact same steps of Appendix A.1, we obtain by replacing the deterministic gradient $\nabla f(x_n)$ by its stochastic estimate $\nabla f(x_n, \xi_{n+1})$ :

$$
\begin{aligned}
H_{n+1} \leq H_n &- \langle a_{n+1}p_{n+1}^2, 1 - \frac{a_{n+1}L}{2} - \frac{|b-(1-\alpha)|}{2u} - \frac{1-\alpha}{2b} \rangle \\
&- \frac{b}{2} \langle a_{n+1}(\nabla f(x_n, \xi_{n+1}) - p_n)^2, \left(1 - \frac{|b-(1-\alpha)|u}{b} - (1-\alpha)\right) \mathbf{1} \rangle \\
&+ \langle \nabla f(x_n, \xi_{n+1}) - \nabla F(x_n), a_{n+1}p_{n+1} \rangle .
\end{aligned} \tag{18}
$$

Using the classical inequality $xy \leq \frac{x^2}{2} + \frac{y^2}{2}$ and the almost sure boundedness of the step size $a_{n+1}$, we get :

$$
\begin{aligned}
\langle \nabla f(x_n, \xi_{n+1}) - \nabla F(x_n), a_{n+1}p_{n+1} \rangle &\leq \langle \frac{1}{2}(\nabla f(x_n, \xi_{n+1}) - \nabla F(x_n))^2 + \frac{1}{2}p_{n+1}^2, a_{n+1} \rangle \\
&\leq \frac{a_{\sup}}{2} \|\nabla f(x_n, \xi_{n+1}) - \nabla F(x_n)\|^2 + \frac{1}{2}\langle a_{n+1}, p_{n+1}^2 \rangle .
\end{aligned}
$$

Therefore, taking the expectation and using the boundedness of the variance, we obtain from Equation (18) :

$$
\mathbb{E}[H_{n+1}] - \mathbb{E}[H_n] \leq -\mathbb{E}\left[ \langle a_{n+1}p_{n+1}^2, 1 - \frac{a_{n+1}L}{2} - \frac{|b-(1-\alpha)|}{2u} - \frac{1-\alpha}{2b} \rangle \right] + \frac{a_{\sup}\sigma^2}{2} .
$$

Then, the proof follows the lines of Appendix A.2. Hence, we have

$$
\mathbb{E}[H_{n+1}] - \mathbb{E}[H_n] \leq -\mathbb{E}\left[ \langle a_{n+1}p_{n+1}^2, \varepsilon\mathbf{1} \rangle \right] + \frac{a_{\sup}\sigma^2}{2} .
$$

We sum these inequalities for $k = 0, \cdots, n-1$ and rearrange the terms to obtain

$$
\mathbb{E}\left[ \sum_{k=0}^{n-1} \langle a_{k+1}, p_{k+1}^2 \rangle \right] \leq \frac{H_0 - \inf f}{\varepsilon} + \frac{n a_{\sup}\sigma^2}{2\varepsilon} .
$$

Then following the derivations in Appendix A.2 using $\nabla f(x_k, \xi_{k+1}) = \frac{1}{b}p_{k+1} - \frac{1-b}{b}p_k$, we establish the following inequality

$$
\mathbb{E}\left[ \sum_{k=0}^{n-1} \langle a_{k+1}, \nabla f(x_k, \xi_{k+1})^2 \rangle \right] \leq \frac{4}{b^2\alpha}\left( \frac{H_0 - \inf f}{\varepsilon} + \langle a_0, p_0^2 \rangle + \frac{n a_{\sup}\sigma^2}{2\varepsilon} \right) .
$$

Finally, we apply Jensen's inequality to $\|\cdot\|^2$, we inject the assumption $a_{n+1} \geq \delta$ and we divide the previous inequality by $n$ to obtain the sought result

$$
\frac{1}{n}\sum_{k=0}^{n-1} \mathbb{E}\left[ \|\nabla F(x_k)\|^2 \right] \leq \frac{4}{n\delta b^2\alpha}\left( \frac{H_0 - \inf f}{\varepsilon} + \langle a_0, p_0^2 \rangle + \frac{n a_{\sup}\sigma^2}{2\varepsilon} \right) .
$$

# B APPENDIX B

## B.1 CONDITIONS SIMILAR TO "GRADIENT-LIKE DESCENT SEQUENCES" CONDITIONS (BOLTE ET AL., 2018, APPENDIX)(OCHS ET AL., 2014)

**Lemma B.1.** Let $(z_k)_{k\in\mathbb{N}}$ be the sequence defined for all $k \in \mathbb{N}$ by $z_k = (x_k, y_k)$ where $y_k = \sqrt{a_k}p_k$ and $(x_k, p_k)$ is generated by Algorithm (2) from a starting point $z_0$. Let Assumptions 2.1 and 4.1 hold true. Assume moreover that condition (6) holds. Then,

(i) (sufficient decrease property) There exists a positive scalar $\rho_1$ s.t. :

$$H(z_{k+1}) - H(z_k) \leq -\rho_1 \|x_{k+1} - x_k\|^2 \quad \forall k \in \mathbb{N}.$$

(ii) There exists a positive scalar $\rho_2$ s.t. :

$$\|\nabla H(z_{k+1})\| \leq \rho_2 \left(\|x_{k+1} - x_k\| + \|x_k - x_{k-1}\|\right) \quad \forall k \geq 1.$$

(iii) (continuity condition) If $\bar{z}$ is a limit point of a subsequence $(z_{k_j})_{j\in\mathbb{N}}$, then $\lim_{j\to+\infty} H(z_{k_j}) = H(\bar{z})$.

**Remark 1.** Conditions in Bolte et al. (2018, Appendix) corresponding to Lemma B.1 are more general. Authors consider a nonsmooth objective function and introduce the Fréchet subdifferential instead of the gradient. For our purposes, this is sufficient.

*Proof.* (i) From Lemma 4.1 and theorem 4.2, we get for all $k \in \mathbb{N}$:

$$H(z_{k+1}) - H(z_k) \leq -\varepsilon\langle a_{k+1}, p_{k+1}^2 \rangle \leq -\varepsilon\left\langle a_{k+1}, \left(\frac{x_{k+1} - x_k}{-a_{k+1}}\right)^2\right\rangle \leq -\frac{\varepsilon}{a_{\sup}} \|x_{k+1} - x_k\|^2.$$

We set $\rho_1 := \frac{\varepsilon}{a_{\sup}}$.

(ii) First, observe that for all $k \in \mathbb{N}$

$$\|\nabla H(z_{k+1})\| \leq \|\nabla f(x_{k+1})\| + \frac{1}{b}\|y_{k+1}\|. \tag{19}$$

Now, let us upperbound each one of these two terms. Recall that we can rewrite our algorithm under a "Heavy-ball"-like form as follows:

$$x_{k+1} = x_k - \alpha_k \nabla f(x_k) + \beta_k(x_k - x_{k-1}) \quad \forall k \geq 1.$$

where $\alpha_k := ba_{k+1}$ and $\beta_k = (1-b)\frac{a_{k+1}}{a_k}$ are vectors.

On the one hand,

$$\begin{aligned}
\|\nabla f(x_{k+1})\|^2 &\leq 2\left(\|\nabla f(x_{k+1}) - \nabla f(x_k)\|^2 + \|\nabla f(x_k)\|^2\right) \\
&\leq 2\left(L^2\|x_{k+1} - x_k\|^2 + \|\nabla f(x_k)\|^2\right) \quad \text{(L-Lipschitz continuity of the gradient)}
\end{aligned}$$

Moreover,

$$\begin{aligned}
\|\nabla f(x_k)\|^2 &= \left\|\frac{x_k - x_{k+1}}{\alpha_k} + \frac{\beta_k}{\alpha_k}(x_k - x_{k-1})\right\|^2 \\
&\leq 2\left\|\frac{x_k - x_{k+1}}{ba_{k+1}}\right\|^2 + 2\left\|\frac{1-b}{b}\frac{1}{a_k}(x_k - x_{k-1})\right\|^2 \\
&\leq \frac{2}{b^2\delta^2}\|x_{k+1} - x_k\|^2 + \frac{2(1-b)^2}{b^2\delta^2}\|x_k - x_{k-1}\|^2 \\
&\leq \frac{2}{b^2\delta^2}\left(\|x_{k+1} - x_k\|^2 + \|x_k - x_{k-1}\|^2\right).
\end{aligned}$$

Hence,

$$\|\nabla f(x_{k+1})\|^2 \leq 2 \left( L^2 \|x_{k+1} - x_k\|^2 + \|\nabla f(x_k)\|^2 \right)$$

$$\leq 2 \left( L^2 + \frac{2}{b^2\delta^2} \right) \|x_{k+1} - x_k\|^2 + \frac{4}{b^2\delta^2} \|x_k - x_{k-1}\|^2$$

$$\leq 2 \left( L^2 + \frac{2}{b^2\delta^2} \right) (\|x_{k+1} - x_k\|^2 + \|x_k - x_{k-1}\|^2) .$$

Therefore, the following inequality holds :

$$\|\nabla f(x_{k+1})\| \leq \sqrt{2 \left( L^2 + \frac{2}{b^2\delta^2} \right)} (\|x_{k+1} - x_k\| + \|x_k - x_{k-1}\|) .$$

On the otherhand,

$$\|y_{k+1}\| = \|\sqrt{a_{k+1}} p_{k+1}\| = \left\| \frac{x_{k+1} - x_k}{\sqrt{a_{k+1}}} \right\| \leq \frac{1}{\sqrt{\delta}} \|x_{k+1} - x_k\| .$$

Finally, combining the inequalities for both terms in Equation (19), we obtain

$$\|\nabla H(z_{k+1})\| \leq \rho_2(\|x_{k+1} - x_k\| + \|x_k - x_{k-1}\|) \quad \forall k \geq 1 .$$

with $\rho_2 := \left( \sqrt{2 \left( L^2 + \frac{2}{b^2\delta^2} \right)} + \frac{1}{b\sqrt{\delta}} \right)$.

(iii) This is a consequence of the continuity of $H$.

$\square$

## B.2 PROOF OF LEMMA 5.1

(i) By Theorem 4.2, the sequence $(H(z_n))_{n\in\mathbb{N}}$ is nonincreasing. Therefore, for all $n \in \mathbb{N}$, $H(z_n) \leq H(z_0)$ and hence $z_n \in \{z : H(z) \leq H(z_0)\}$. Since $f$ is coercive, $H$ is also coercive and its level sets are bounded. As a consequence, $(z_n)_{n\in\mathbb{N}}$ is bounded and there exist $z_* \in \mathbb{R}^d$ and a subsequence $(z_{k_j})_{j\in\mathbb{N}}$ s.t. $z_{k_j} \to z_*$ as $j \to \infty$. Hence, $\omega(z_0) \neq \emptyset$. Furthermore, $\omega(z_0) = \bigcap_{q\in\mathbb{N}} \overline{\bigcup_{k\geq q}\{z_k\}}$ is compact as an intersection of compact sets.

(ii) First, $\text{crit}H = \text{crit}f \times \{0\}$ because $\nabla H(z) = (\nabla f(x), y/b)^T$. Let $z_* \in \omega(z_0)$. Recall that $x_{k+1} - x_k \to 0$ as $k \to \infty$ by Theorem 4.2. We deduce from the second assertion of Lemma B.1 that $\nabla H(z_k) \to 0$ as $k \to \infty$. As $z_* \in \omega(z_0)$, there exists a subsequence $(z_{k_j})_{j\in\mathbb{N}}$ converging to $z_*$. Then, by Lipschitz continuity of $\nabla H$, we get that $\nabla H(z_{k_j}) \to \nabla H(z_*)$ as $j \to \infty$. Finally, $\nabla H(z_*) = 0$ since $\nabla H(z_k) \to 0$ and $(\nabla H(z_{k_j}))_{j\in\mathbb{N}}$ is a subsequence of $(\nabla H(z_n))_{n\in\mathbb{N}}$ .

(iii) This point stems from the definition of limit points. Every subsequence of the sequence $(\text{d}(z_k, \omega(z_0)))_{k\in\mathbb{N}}$ converges to zero as a consequence of the definition of $\omega(z_0)$.

(iv) The sequence $(H(z_n))_{n\in\mathbb{N}}$ is nonincreasing by Theorem 4.2. It is also bounded from below because $H(z_k) \geq f(x_k) \geq \inf f$ for all $k \in \mathbb{N}$. Hence we can denote by $l$ its limit. Let $\bar{z} \in \omega(z_0)$. There there exists a subsequence $(z_{k_j})_{j\in\mathbb{N}}$ converging to $\bar{z}$ as $j \to \infty$. By the third assertion of Lemma B.1, $\lim_{j\to+\infty} H(z_{k_j}) = H(\bar{z})$. Hence this limit equals $l$ since $(H(z_n))_{n\in\mathbb{N}}$ converges towards $l$. Therefore, the restriction of $H$ to $\omega(z_0)$ equals $l$ .

## B.3 PROOF OF THEOREM 5.3

The first step of this proof follows the same path as Bolte et al. (2018, Proof of Theorem 6.2, Appendix). Since $f$ is coercive, $H$ is also coercive. The sequence $(H(z_k))_{k\in\mathbb{N}}$ is nonincreasing. Hence, $(z_k)$ is bounded and there exists a subsequence $(z_{k_q})_{q\in\mathbb{N}}$ and $\bar{z} \in \mathbb{R}^{2d}$ s.t. $z_{k_q} \to \bar{z}$ as $q \to \infty$. Then, since $(H(z_k))_{k\in\mathbb{N}}$ is nonincreasing and lowerbounded by $\inf f$, it is convergent and we obtain by continuity of $H$,

$$\lim_{k\to+\infty} H(z_k) = H(\bar{z}) . \quad (20)$$

If there exists $\bar{k} \in \mathbb{N}$ s.t. $H(z_{\bar{k}}) = H(\bar{z})$, then $H(z_{\bar{k}+1}) = H(\bar{z})$ and by the first point of Lemma B.1, $x_{\bar{k}+1} = x_{\bar{k}}$ and then $(x_k)_{k \in \mathbb{N}}$ is stationary and for all $k \geq \bar{k}$, $H(z_k) = H(\bar{z})$ and the results of the theorem hold in this case (note that $\bar{z} \in \text{crit} H$ by Lemma 5.1). Therefore, we can assume now that $H(\bar{z}) < H(z_k) \forall k > 0$ since $(H(z_k))_{k \in \mathbb{N}}$ is nonincreasing and Equation (20) holds. One more time, from Equation (20), we have that for all $\eta > 0$, there exists $k_0 \in \mathbb{N}$ s.t. $H(z_k) < H(\bar{z}) + \eta$ for all $k > k_0$. From Lemma 5.1, we get $\text{d}(z_k, \omega(z_0)) \to 0$ as $k \to +\infty$. Hence, for all $\varepsilon > 0$, there exists $k_1 \in \mathbb{N}$ s.t. $\text{d}(z_k, \omega(z_0)) < \varepsilon$ for all $k > k_1$. Moreover, $\omega(z_0)$ is a nonempty compact set and $H$ is finite and constant on it. Therefore, we can apply the uniformization Lemma 5.2 with $\Omega = \omega(z_0)$. Hence, for any $k > l := \max(k_0, k_1)$, we get

$$\varphi'(H(z_k) - H(\bar{z}))^2 \|\nabla H(z_k)\|^2 \geq 1. \tag{21}$$

This completes the first step of the proof. In the second step, we follow the proof of Johnstone and Moulin (2017, Theorem 2). Using Lemma B.1 .(i)-(ii), we can write for all $k \geq 1$,

$$\|\nabla H(z_{k+1})\|^2 \leq 2\rho_2^2 \left(\|x_{k+1} - x_k\|^2 + \|x_k - x_{k-1}\|^2\right) \leq \frac{2\rho_2^2}{\rho_1}(H(z_{k-1}) - H(z_{k+1})).$$

Injecting the last inequality in Equation (21), we obtain for all $k > k_2 := \max(l, 2)$,

$$\frac{2\rho_2^2}{\rho_1} \varphi'(H(z_k) - H(\bar{z}))^2 (H(z_{k-2}) - H(z_k)) \geq 1.$$

Now, use $\varphi'(s) = \bar{c}s^{\theta-1}$ to derive the following for all $k > k_2$:

$$[H(z_{k-2}) - H(\bar{z})] - [H(z_k) - H(\bar{z})] \geq \frac{\rho_1}{2\rho_2^2 \bar{c}^2}[H(z_k) - H(\bar{z})]^{2(1-\theta)}. \tag{22}$$

Let $r_k := H(z_k) - H(\bar{z})$ and $C_1 = \frac{\rho_1}{2\rho_2^2 \bar{c}^2}$. Then, we can rewrite Equation (22) as

$$r_{k-2} - r_k \geq C_1 r_k^{2(1-\theta)} \quad \forall k > k_2. \tag{23}$$

We distinguish three different cases to obtain the sought results.

(i) $\underline{\theta = 1}$:
Suppose $r_k > 0$ for all $k > k_2$. Then, since we know that $r_k \to 0$ by Equation (20), $C_1$ must be equal to 0. This is a contradiction. Therefore, there exist $k_3 \in \mathbb{N}$ s.t. $r_k = 0$ for all $k > k_3$ (recall that $(r_k)_{k \in \mathbb{N}}$ is nonincreasing).

(ii) $\underline{\theta \geq \frac{1}{2}}$:
As $r_k \to 0$, there exists $k_4 \in \mathbb{N}$ s.t. for all $k \geq k_4$, $r_k \leq 1$. Observe that $2(1-\theta) \leq 1$ and hence $r_{k-2} - r_k \geq C_1 r_k$ for all $k > k_2$ and then

$$r_k \leq (1 + C_1)^{-1} r_{k-2} \leq (1 + C_1)^{-p_1} r_{k_4}. \tag{24}$$

where $p_1 := \lfloor \frac{k-k_4}{2} \rfloor$. Notice that $p_1 > \frac{k-k_4-2}{2}$. Thus, the linear convergence result follows. Note also that if $\theta = 1/2$, $2(1-\theta) = 1$ and Equation (24) holds for all $k > k_2$.

(iii) $\underline{\theta < \frac{1}{2}}$:
Define the function $h$ by $h(t) = \frac{D}{1-2\theta}t^{2\theta-1}$ where $D > 0$ is a constant. Then,

$$h(r_k) - h(r_{k-2}) = \int_{r_{k-2}}^{r_k} h'(t)dt = D \int_{r_k}^{r_{k-2}} t^{2\theta-2}dt \geq D(r_{k-2} - r_k) r_{k-2}^{2\theta-2}.$$

We disentangle now two cases :

(a) Suppose $2r_{k-2}^{2\theta-2} \geq r_k^{2\theta-2}$. Then, by Equation (23), we get

$$h(r_k) - h(r_{k-2}) = D(r_{k-2} - r_k) r_{k-2}^{2\theta-2} \geq \frac{C_1 D}{2}. \tag{25}$$

(b) Suppose now the opposite inequation $2r_{k-2}^{2\theta-2} < r_k^{2\theta-2}$. We can suppose without loss of generality that $r_k$ are all positive. Otherwise, if there exists $p$ such that $r_p = 0$, the sequence $(r_k)_{k\in\mathbb{N}}$ will be stationary at $0$ for all $k \geq p$. Observe that $2\theta - 2 < 2\theta - 1 < 0$, thus $\frac{2\theta-1}{2\theta-2} > 0$. As a consequence, we can write in this case $r_k^{2\theta-1} > q\, r_{k-2}^{2\theta-1}$ where $q := 2^{\frac{2\theta-1}{2\theta-2}} > 1$. Therefore, using moreover that the sequence $(r_k)_{k\in\mathbb{N}}$ is nonincreasing and $2\theta - 1 < 0$, we derive the following

$$h(r_k) - h(r_{k-2}) = \frac{D}{1-2\theta}(r_k^{2\theta-1} - r_{k-2}^{2\theta-1}) > \frac{D}{1-2\theta}\,(q-1)r_{k-2}^{2\theta-1} > \frac{D}{1-2\theta}\,(q-1)r_{k_2}^{2\theta-1} := C_2\,.$$

(26)

Combining Equation (25) and Equation (26) yields $h(r_k) \geq h(r_{k-2}) + C_3$ where $C_3 := \min(C_2, \frac{C_1 D}{2})$. Consequently, $h(r_k) \geq h(r_{k-2\,p_2}) + p_2\, C_3$ where $p_2 := \lfloor \frac{k-k_2}{2} \rfloor$. We deduce from this inequality that

$$h(r_k) \geq h(r_k) - h(r_{k-2\,p_2}) \geq p_2\, C_3\,.$$

Therefore, rearranging this inequality using the definition of $h$, we obtain $r_k^{1-2\theta} \leq \frac{D}{1-2\theta}(C_3\, p_2)^{-1}$. Then, since $p_2 > \frac{k-k_2-2}{2}$,

$$r_k \leq C_4\, p_2^{\frac{1}{2\theta-1}} \leq C_4 \left( \frac{k-k_2-2}{2} \right)^{\frac{1}{2\theta-1}}\,.$$

where $C_4 := \left( \frac{C_3\,(1-2\theta)}{D} \right)^{\frac{1}{2\theta-1}}$.

We conclude the proof by observing that $f(x_k) \leq H(z_k)$ and recalling that $\bar{z} \in \mathrm{crit}H$.

## B.4 PROOF OF LEMMA 5.4

Since $f$ has the KŁ property at $\bar{x}$ with an exponent $\theta \in (0, 1/2]$, there exist $c, \varepsilon$ and $\nu > 0$ s.t.

$$\|\nabla f(x)\|^{\frac{1}{1-\theta}} \geq c(f(x) - f(\bar{x}))$$

(27)

for all $x \in \mathbb{R}^d$ s.t. $\|x - \bar{x}\| \leq \varepsilon$ and $f(x) < f(\bar{x}) + \nu$ where condition $f(\bar{x}) - f(x)$ is dropped because Equation (27) holds trivially otherwise. Let $z = (x, y) \in \mathbb{R}^{2d}$ be s.t. $\|x - \bar{x}\| \leq \varepsilon$, $\|y\| \leq \varepsilon$ and $H(\bar{x}, 0) < H(x, y) < H(\bar{x}, 0) + \nu$. We assume that $\varepsilon < b$ ($\varepsilon$ can be shrunk if needed). We have $f(x) \leq H(x, y) < H(\bar{x}, 0) + \nu = f(\bar{x}) + \nu$. Hence Equation (27) holds for these $x$.

By concavity of $u \mapsto u^{\frac{1}{2(1-\theta)}}$, we obtain

$$\|\nabla H(x, y)\|^{\frac{1}{1-\theta}} \geq C_0 \left( \|\nabla f(x)\|^{\frac{1}{1-\theta}} + \left\| \frac{y}{b} \right\|^{\frac{1}{1-\theta}} \right)$$

where $C_0 := 2^{\frac{1}{2(1-\theta)}-1}$.

Hence, using Equation (27), we get

$$\|\nabla H(x, y)\|^{\frac{1}{1-\theta}} \geq C_0 \left( c\,(f(x) - f(\bar{x})) + \left\| \frac{y}{b} \right\|^{\frac{1}{1-\theta}} \right)\,.$$

Observe now that $\frac{1}{1-\theta} \geq 2$ and $\left\| \frac{y}{b} \right\| \leq \frac{\varepsilon}{b} \leq 1$. Therefore, $\left\| \frac{y}{b} \right\|^{\frac{1}{1-\theta}} \geq \|y/b\|^2$.

Finally,

$$\|\nabla H(x, y)\|^{\frac{1}{1-\theta}} \geq C_0 \left( c\,(f(x) - f(\bar{x})) + \frac{2}{b}\frac{1}{2b}\|y\|^2 \right)$$

$$\geq C_0 \min\left( c, \frac{2}{b} \right) \left( f(x) - f(\bar{x}) + \frac{1}{2b}\|y\|^2 \right)$$

$$= C_0 \min\left( c, \frac{2}{b} \right) (H(x, y) - H(\bar{x}, 0))\,.$$

This completes the proof.

### B.5 COMPARISON TO OCHS ET AL. (2014)

We recall the conditions satisfied by $\alpha_n$ and $\beta_n$ in Ochs et al. (2014) in order to traduce them in terms of the algorithm (2) at stake. Define :

$$\delta_n := \frac{1}{\alpha_n} - \frac{L}{2} - \frac{\beta_n}{2\alpha_n} \qquad \gamma_n := \delta_n - \frac{\beta_n}{2\alpha_n}.$$

Conditions of Ochs et al. (2014) write: $\alpha_n \geq c_1 \; \beta_n \geq 0 \; \delta_n \geq \gamma_n \geq c_2$ where $c_1, c_2$ are positive constants and $(\delta_n)$ is monotonically decreasing.

One can remark that algorithm (2) can be written as (4) with step sizes $\alpha_n = ba_{n+1}$ and inertial parameters $\beta_n = (1 - b)\frac{a_{n+1}}{a_n}$. Conditions on these parameters can be expressed in terms of $a_n$. Supposing $c_2 = 0$, the condition $\gamma_n \geq c_2$ is equivalent to

$$\frac{a_{n+1}}{a_n} \leq \frac{2}{2 - b(2 - a_n L)}. \tag{28}$$

Note that the classical condition $a_n \leq 2/L$ shows up consequently. Moreover, the condition on $(\delta_n)$ is equivalent to

$$\frac{1}{a_{n+1}} \leq \frac{3 - b}{2} \frac{1}{a_n} - \frac{1 - b}{2a_{n-1}} \qquad \text{for} \qquad n \geq 1. \tag{29}$$

Note that we get rid of condition (29) while allowing adaptive step sizes $a_n$ (see Proposition A.1).

### B.6 A CONVERGENCE RESULT FOR GRADIENT DESCENT IN THE NONCONVEX SETTING.

Consider the gradient descent algorithm defined by : $x_{k+1} = x_k - \gamma \nabla f(x_k)$. Assume that $\gamma > 0$ and $1 - \frac{\gamma L}{2} > 0$.

Supposing that $\nabla f$ is $L-$Lipschitz, using Taylor's expansion and regrouping the terms, we obtain the following inequality:

$$f(x_{k+1}) \leq f(x_k) - \gamma \left( 1 - \frac{\gamma L}{2} \right) \|\nabla f(x_k)\|_2^2.$$

Then, we sum the inequalities for $0 \leq k \leq n - 1$, lower bound the gradients norms in the sum by their minimum and we obtain :

$$\min_{0 \leq k \leq n-1} \|\nabla f(x_k)\|_2^2 \leq \frac{f(x_0) - \inf f}{n\gamma(1 - \frac{\gamma L}{2})}.$$

### B.7 ABOUT THE KŁ-PROPERTY

**A simple example for more intuition.** For building intuition, let us consider the 1D function $g$ defined by $g(x) = |x|^p$ for $p \geq 2$ as exposed in Johnstone and Moulin (2017). The function $\varphi(t) = t^{1/p}$ is a desingularizing function. When $g$ is flat around the origin (large $p$), gradient methods are slower to converge. Therefore, a smaller KŁ exponent encodes a slower convergence behavior near a critical point.

**Remark 2.** Definition 5.2 can be generalized to a nonsmooth function by introducing the notion of the Fréchet subdifferential (which is a set) and replacing the norm by the distance to a set.

