# OpenReview forum: "Convergence Analysis of a Momentum Algorithm with Adaptive Step Size for Nonconvex Optimization"
_ICLR.cc/2020/Conference — Reject_

### Official Review · AnonReviewer1 · 2019-10-23
**Official Blind Review #1**

**Rating:** 3

**Review:**

This paper provides convergence analyses for momentum methods using adaptive step size for non-convex problems under a bounded assumption on learning rates. Concretely, a sublinear convergence rate under a general setting and improved convergence rates under KL-condition are provided.

An interesting point of the paper is that (i) the boundedness assumption on the domain or gradient is not required and that (ii) shows faster convergence rates under KL-condition for non-descent methods. However, a major concern is a uniform boundedness assumption on learning rates. A lower bound is a matter because this condition is verified a posteriori after running methods. This kind of condition is not preferred in general. Indeed, most studies provide convergence analyses without such assumptions. In addition, there are several missing references (listed below) that attempt to analyze the convergence of adaptive methods. To make the position of the paper clear, it would be better to provide a theoretical comparison with these studies. Especially, [LO2019] and [XWW2019] are related to this study. [LO2019] has provided convergence analyses without boundedness assumptions on the domain or gradient and [XWW2019] has provided a linear or better convergence rate of an adaptive method by utilizing the KL-condition. Note that a method treated in [XWW2019] is also a non-descent method because there is no theoretical limitation on the initial step size.

[WWB2018] X.Wu, R.Ward, and L.Bottou, WNGrad: Learn the Learning Rate in Gradient Descent. arXiv, 2018.
[WWB2019] R.Ward, X.Wu, and L.Bottou. AdaGrad Stepsizes: Sharp Convergence Over Nonconvex Landscapes. ICML, 2019.
[XWW2019] Y.Xie, X.Wu, and R.Ward. Linear Convergence of Adaptive Stochastic Gradient Descent. arXiv, 2019.
[Levy2017] K.Y.Levy, Online to Offline Conversions, Universality and Adaptive Minibatch Sizes. NIPS, 2017.
[LYC2018] Y.K.Levy, A.Yurtsever, and V.Cevher, Online Adaptive Methods, Universality and Acceleration, NeurIPS, 2018.
[LO2019] X.Li, and F.Orabona. On the Convergence of Stochastic Gradient Descent with Adaptive Stepsizes. AISTATS, 2019.




**Experience Assessment:**

I have read many papers in this area.

**Review Assessment: Checking Correctness Of Derivations And Theory:**

I did not assess the derivations or theory.

**Review Assessment: Checking Correctness Of Experiments:**

N/A

**Review Assessment: Thoroughness In Paper Reading:**

N/A

---

### Official Review · AnonReviewer2 · 2019-10-23
**Official Blind Review #2**

**Rating:** 3

**Review:**

In this work, the authors consider a variation of ADAM with a boundedness assumption on the step size and focus on unconstrained, smooth, non-convex minimization setting.

The authors provide non-asymptotic convergence rates for deterministic and stochastic settings, with respect to the gradient norm. Moreover, they also prove convergence in function value for a class of functions that satisfy Kurdyka-Lojasiewicz (KL) property. To the best of my knowledge, this is the first work that utilizes KL property and respective analysis for ADAM-like methods. I have to note that I disagree with your statement of being “adaptive”.

I summarize my comments step by step below:

I find the presentation and organization of the paper clear and structured. Related work is sufficient, the authors cite relevant papers with respect to convergence in deterministic and stochastic setting and present detailed comparison between their framework. Similarly, the momentum idea is motivated through Polyak’s Heavy ball. The known rates for KL functions are provided in conjunction with momentum-based methods.

Zaheer et al. (2018) assume \beta_1 = 0, meaning their proof works for RMSProp. However for this paper, given that Theorems 4.2 and 4.3 are provided for Algorithm (2), then the algorithm in question is not exactly the same as ADAM either, in my opinion. The bias correction steps are missing. The authors should clarify this point in the paper.

Definition of adaptivity is a little vague for me. There exist different notions of adaptivity (adaptation to global/local smoothness without knowing L, adaptation to non-smooth & smooth problems simultaneously etc.) The authors define adaptivity as “computing step size using gradient history”. However, their upper bound for the step size requires knowledge of L. Also, they assume a uniform lower bound for step size. These make their framework rather non-adaptive in my opinion.

Follow up comment: How would one apply this method to neural networks, or any class of problems where Lipschitz constant is virtually unknown?

Convergence rate in deterministic oracle setting, provided in Theorem 4.2, is plausible. The proof is nice and easy to read. Compared to De et al., the authors show faster rates with respect to number of iterations. But the rate depends on upper/lower bounds of the step size.

Convergence rates in the stochastic setting suggest the quantity does not go to zero, but converges to some noise dominated region (equivalently to some neighborhood) of stationary points. Zaheer et al. has a similar rate characterization for RMSProp, but this paper considers first order moment accumulation on top Zaheer et al. I am not sure about the impact of this results compared to existing ones for ADAM variants. Similar to deterministic case, the analysis is nice and clean. The rate has no dependence on dimension, but so does the analysis of De et al. (2018) and Zaheer et al. (2018). It is not new, but it is a desirable property of the analysis.

I have doubts about “not having bounded gradient assumption”. There exists a uniform lower bound for a_n, which implies the sum of square of each coordinate of gradients are bounded, which implies the infinity norm of gradients is bounded. I believe there is an implicit assumption of bounded gradients. It also makes me question if somehow the dimension dependence is hidden under this step size lower bound. I would expect the authors to address the concerns about bounded gradient and dimension dependence.

Convergence characterization for KL functions with (sort of) adaptive step size is new also to the best of my knowledge. The convergence analysis is based on PALM (Bolte et al. (2014)) and Bolte et al. (2018) and these previous results are adapted to ADAM. My concern about this direction is whether neural networks belong to this class of functions. If not, how useful it is to use ADAM for this class of functions?

Overall, authors approach ADAM-like methods from a generalized scheme based on Heavy Ball, and provide analysis for smooth, non-convex functions with deterministic/stochastic gradients. Having dimension-independent rates is a positive trait of the analysis, but I would like to see a clarification regarding my previous point. In the stochastic setting, dimension-free rate for ADAM (which has very similar characterization to RMSProp in Zaheer et al.) seems to be an important results of this paper. I think the KL function analysis is a new result for “adaptive”, evolving step sizes, as well. However, I do not agree with the claim that the authors do not make bounded gradient assumption. In a way, the uniform lower bound on the step size implies it. Knowledge of Lipschitz constant (through step size upper bound) is a restriction for “highly non-convex” neural network optimization problems. Considering that ADAM is a classical optimizer for neural networks, I also doubt KL property holds for complex networks. Also, I am not convinced that the algorithm is truly adaptive (upper bounding step size with a function of Lipschitz constant.)

I am inclined to give a weak reject to this paper because I am not convinced that the results presented in the paper are compatible with the application. I would also like to see the justification/clarification of the authors for my previous concerns before making the final decision.

**Experience Assessment:**

I have read many papers in this area.

**Review Assessment: Checking Correctness Of Derivations And Theory:**

I assessed the sensibility of the derivations and theory.

**Review Assessment: Checking Correctness Of Experiments:**

N/A

**Review Assessment: Thoroughness In Paper Reading:**

I read the paper at least twice and used my best judgement in assessing the paper.

---

### Official Review · AnonReviewer3 · 2019-10-23
**Official Blind Review #3**

**Rating:** 3

**Review:**

This work analyzes the performance of ADAM algorithm under a bounded step size assumption. it proves convergence rate for the deterministic setting and prove a convergence result for the stochastic setting.


However, I have a few concerns as follows. The first two comments are the major reasons of my rating.
1. the convergence in Theorem 4.2 is not standard. It is about "the minimum of the gradients norms", which does not imply the algorithmic convergence. it should be more clearly stated in the introduction.
2. The author should include simulations to show whether the theoretical result matches the empirical performance, for both deterministic setting and stochastic setting.

3. Small comment: In section 2, g_i should be a d-dimensional vector since it is the gradient of f. But then what is g_i^2?
4. For comparison, can you set b=1 and compare with the existing theoretical results of RMSPROP?


**Experience Assessment:**

I have read many papers in this area.

**Review Assessment: Checking Correctness Of Derivations And Theory:**

I assessed the sensibility of the derivations and theory.

**Review Assessment: Checking Correctness Of Experiments:**

N/A

**Review Assessment: Thoroughness In Paper Reading:**

I read the paper at least twice and used my best judgement in assessing the paper.

---

### Decision · Program_Chairs · 2019-12-19

**Decision:**

Reject

**Comment:**

The reviewers have reached consensus that while the paper is interesting, it could use more time.  We urge the authors to continue their investigations.